# Origin of Genome Instability and Determinants of Mutational Landscape in Cancer Cells

**DOI:** 10.3390/genes11091101

**Published:** 2020-09-21

**Authors:** Sonam Mehrotra, Indraneel Mittra

**Affiliations:** Advanced Centre for Treatment, Research and Education in Cancer (ACTREC), Tata Memorial Centre, Kharghar, Navi Mumbai 410210, India; imittra@actrec.gov.in

**Keywords:** genome instability, replication stress, replication-transcription conflict, cell free chromatin, cancer

## Abstract

Genome instability is a crucial and early event associated with an increased predisposition to tumor formation. In the absence of any exogenous agent, a single human cell is subjected to about 70,000 DNA lesions each day. It has now been shown that physiological cellular processes including DNA transactions during DNA replication and transcription contribute to DNA damage and induce DNA damage responses in the cell. These processes are also influenced by the three dimensional-chromatin architecture and epigenetic regulation which are altered during the malignant transformation of cells. In this review, we have discussed recent insights about how replication stress, oncogene activation, chromatin dynamics, and the illegitimate recombination of cell-free chromatin particles deregulate cellular processes in cancer cells and contribute to their evolution. The characterization of such endogenous sources of genome instability in cancer cells can be exploited for the development of new biomarkers and more effective therapies for cancer treatment.

## 1. Introduction

Genome instability is a characteristic feature observed in most human cancers. The accumulation of genetic alterations ranging from single nucleotide mutations to chromosome rearrangements can predispose cells towards malignancy. In addition, sustained genome instability enables cancer cells to survive under selective pressure and adapt to their microenvironments by evolving mechanisms to resist different therapies [1]. High throughput sequencing efforts, including those by the International Cancer Genome Consortium (ICGC) and The Cancer Genome Atlas (TCGA), have identified and documented key driver mutations associated with different types of cancers. Primarily, they comprise of mutations in DNA repair genes, tumor suppressor genes, and oncogenes [2,3]. The Pan-Cancer Analysis of Whole Genome (PNAWG) Consortium has systematically characterized a repertoire of mutational signatures and their contribution to the development of different human cancers. These analyses identified single, double, or cluster base substitutions, small insertions and deletions, copy number changes, and genome rearrangements. These data also suggest that genomes of individual cancer cells exhibit multiple superimposed mutational signatures which are possibly generated by multiple processes involving DNA replication, modification, damage, and damage response [4]. The association of these mutational signatures and the timing of their acquisition with cancer phenotype, incidence, and etiology is not clearly understood [5]. Hence, it is imperative to understand different mechanisms that shape the mutational landscape in cancer cells during different stages of tumorigenesis and how they impact prognosis as well as treatment strategies.

It has been estimated that a single nucleated human cell is subjected to approximately 70,000 DNA lesions each day, out of which 10–50 lesions are DNA double strand breaks (DSBs). Although DSBs occur at a much lower frequency compared to other DNA lesions, they constitute the most detrimental of DNA lesions and imperil genome integrity [6]. If not faithfully repaired, DSBs can result in mutations, chromosomal translocations, or rearrangements associated with cancer, diabetes, and other disorders [1]. Although exogenous agents such as ionizing radiation, DNA chelating agents, or radiomimetic chemicals induce DNA DSBs, the majority of DSBs are generated due to cellular metabolism and DNA transactions during DNA replication and transcription. The response to DNA damage is regulated by both genetic and epigenetic factors and also influenced by the three dimensional organization of the chromatin [7]. There is emerging evidence suggesting that major changes in the chromatin architecture accompany genome levels changes underlying tumor initiation and metastasis [8].

Recently, a new phenomenon has been discovered which suggests that dying cancer cells release cell free chromatin (cfCh) which is capable of illegitimately integrating into the genome of cells present in the microenvironment both locally and that of distant organs. The integration of cfCh further alters the genomes of these cells and promotes neoplasia [9]. Understanding this novel mechanism is critical to elucidate how different cancer cells respond to therapy and develop resistance.

Cells utilize different DNA repair pathways in coordination with an elaborate network of signaling mechanisms which regulate the cell cycle in presence of DNA lesions to allow repair or initiate cell death in case of irreparable DNA lesions [10]. Cells also possess a robust surveillance mechanism which coordinates with the repair machinery for sensing and repairing DNA damage. DSBs are predominantly repaired by two partially redundant pathways known as the homologous recombination (HR) and the non-homologous end-joining (NHEJ) [11,12]. The HR mediated DNA repair is active during S and G2 phase of the cell cycle and uses equivalent regions of DNA such as the sister chromatid or the homologous chromosome as the template. On the contrary, NHEJ is predominantly active during G1 phase and involves ligation of the two ends of the DNA lesion without any DNA synthesis that employs any equivalent DNA sequence as the template. Hence, the HR mediated DNA repair is a high-fidelity error free mode of DNA repair, whereas NHEJ frequently results in deletions or insertions at the break site and is prone to error [11]. Therefore, DSBs repaired by mechanisms other than HR are an important source of genome alteration that result in genome instability.

It is well established that genome instability arises from defects in high fidelity pathways involved in DNA repair and/or genotoxic stress that originates from cellular processes overwhelming the DNA repair machinery in the cell. The alterations in DNA repair pathways associated with different types of cancer have been extensively studied. However, how DNA lesions resulting from essential DNA transactions, oncogene activation, 3D-chromatin organization, and the integration of illegitimate cfCh shape the genomic landscape of cancer cell remain unclear. In this review, we discuss the role of the aforementioned mechanisms as endogenous sources of genome instability and their contributions towards the evolution of cancer cells.

## 2. Replication Stress as a Source of Endogenous Double Strand Breaks

The precise replication of all chromosomes once every cell cycle is essential for maintaining genome stability in eukaryotic cells. This is a highly conserved phenomenon among different organisms that involves unwinding and duplication of the entire genetic material while also retaining the regulatory epigenetic information [13]. 

Occasionally, the replication fork progression encounters physical impediments such as DNA lesions, unusual DNA structures, conflicts with transcription machinery, or the depletion of key biomolecules required for DNA synthesis; all of which results in slowing down or stalling of progressive replication forks referred to as replication stress. Most stalled forks are capable of resuming their progression in a short time frame after the replication block, however, some may collapse if their replication machinery dissociates from the DNA template [14]. The collapsed replication forks are processed by the action of topoisomerases and structure specific endonuclease complexes in turn lead to the generation of DSBs. The deficiency of these endonuclease complexes like MUS81-EME1 and GEN1 are associated with increased cancer predisposition [15].

Stalled replication forks reverse and can undergo branch migration in the direction opposite to that of the fork progression and are followed by reannealing of the newly synthesized nascent DNA strands. This process results in the formation of Holliday junction-like intermediates which have to be resolved in order to resume the replication of that replisome. The reversal or regression of stalled replication forks are protective mechanisms that ensure their stabilization and prevent collapse. These processes involve several recombinases, DNA nucleases, and helicases, for example, RAD51, RAD52, BRCA1, BRCA2, MRE11, EXO1, SMARCAL1, RECQL5, and BLM [16]. Many of these proteins are also involved in DNA repair mechanisms, where they have mechanistically distinct functions. The process of fork reversal results in generation of ssDNA which binds to RPA proteins. RPA protein together with the 9-1-1 complex and TOPBP1 activates the ATR kinase. The ATR kinase phosphorylates CHK1 that suppresses the firing of new origins and transition into S phase, thus allowing time for DNA damage repair. The complex comprising of the proteins Timeless, Claspin, and Tipin associates with the progressing replication fork. These are required to sense and induce DNA repair during replication [17]. Different DNA lesions are repaired using different DNA repair mechanisms such as the HR pathway, the Fanconi Anaemia Pathway, NHEJ, or post replicative repair. Compromised DNA repair pathways and defective activation of checkpoints lead to unresolved and incomplete replication intermediates which exacerbate genome instability and accelerate tumorigenesis [13,18]. Replication stress thus poses a major challenge to genome integrity. In this section, we discuss some factors that increase replication stress in cancer cells.

### 2.1. Replication Stress Due to DNA Lesions and Unusual DNA Structures

Different types of DNA lesions hinder replication fork progression, causing it to stall in order to enable DNA repair. For example, nucleotide misincorporation by DNA polymerases lead to single base lesions which are repaired by base excision repair (BER) mechanisms. Single strand break repair also results in ssDNA which can promote the dissociation of DNA polymerase at the replication fork. DNA–DNA interstrand or DNA-protein crosslinks are also sufficient to terminate replication fork progression [19]. 

It has been estimated that during DNA replication, about 1 million ribonucleotides can be erroneously incorporated by DNA polymerases. Removal of such ribonucleotides requires RNAse H2 mediated ribonucleotide excision [20]. Deficiency of RNAse H enzymes cause accumulation of ribonucleotides in the DNA sequence which are alternatively removed by the action of DNA topoisomerases mediated mechanisms. These however, lead to generation of ssDNA and DSBs [21]. 

It has been determined that certain human stem cells accumulate approximately 40 novel mutations per year [22,23]. In some instances, DNA synthesis can be re-initiated by bypassing a DNA lesion hindering a progressing replication fork. This type of trans-lesion DNA synthesis is facilitated by DNA polymerase Primpol, which also possesses primase activity. Primpol mediated DNA synthesis is, however, error prone and therefore allows the tolerance or even the transmission of such mutations [24,25]. As described earlier, regions of ssDNA generated as a result of replication stress can also serve as substrates for APOBECs, which are a type of cytosine deaminase enzymes involved in the suppression of retroviruses and other mobile elements [26]. These enzymes are observed to be upregulated in almost 50% of all types of cancers. Thus, ssDNA generated as a result of stalled replication fork reversal increases the mutation burden in cancer cells.

In addition to DNA lesions, certain DNA sequences also have the tendency to form structures comprising of non-B-form DNA conformations. These include G-quadraplexes, Z-form, triplex DNA, and hairpins. These structures are often sufficient in posing physical impediments that can slow down or terminate progressing replication forks [27]. Repeated motifs and DNA sequences such as those found in microsatellite, transposable elements, and terminal repeats are susceptible to DNA damages resulting from the repeat deletion of expansions. These slow replicating regions known as fragile sites are prone to DNA breaks and chromatin breaks during mitosis. Replication perturbations at such fragile sites result in genome and chromosomal instability associated with many cancers [28,29].

### 2.2. Aberrant Origin Licensing and Initiation

The process of DNA replication is tightly regulated by the sequential assembly of a number of protein complexes at defined DNA loci that constitute the origin. This process, known as ‘origin licensing’, is restricted between late mitotic and early G1 phases ensures the initiation of DNA replication only once per cell cycle. A complex of proteins is formed first by the binding of Origin Recognition Complex Proteins (ORC) to the origin loci. ORC binding recruits CDC6, and CDT1 proteins that facilitate the binding of the MCM2-7 DNA helicase and form the pre-replication complex (pre-RC). At this stage, the helicase complex remains inactive and does not begin unwinding of the two DNA strands [30,31,32]. 

With the concerted action of CDC7 kinases and the recruitment of the GIN complex, the MCM DNA helicases are activated and the cells transition into S-phase. This is followed by the formation of the active replisome or a replication bubble which proceeds bidirectionally outward from the origin [33]. During every cell cycle, replication is initiated from only a small number of licensed origins. The excess licensed origins remain dormant and serve as backups in case of replication stress or failure [34]. Mutations in components of the preRC reduce the number of licensed origins which may result in under replicated regions in the genome that are predisposed to DNA breakage and other rearrangements [35]. 

Several mechanisms prevent the unscheduled localization of pre-RC components at origins which in turn prevents re-replication during the cell cycle. Periodic gene expression of CDK and their regulatory cyclins as well as anaphase promoting complex (APC) are key factors involved in the regulation of origin licensing. This process is temporally restricted to a small-time window during late mitosis and early G1, by high APC and low CDK activity. High CDK activity during the S and G2 phases inhibits the re-licensing of origin before passage through mitosis [36]. Other mechanisms that are important in regulation of replication timing include the inhibition of pre-RC proteins by phosphorylation and their ubiquitin mediated degradation [37]. The accumulation of CDT1 and CDC6 proteins alters replication timing due to unscheduled licensing of origins and in turn increase the rate of replication fork collision and collapse leading to DSBs. The overexpression and accumulation of CDT1 CDC6 and other licensing factors are frequently observed in early stage epithelial cancers [38]. 

### 2.3. Oncogene Induced Replication Stress

Re-replication is also frequently observed as a result of the activation of different oncogenes which bring about major changes in the genome [13,39]. Many of the proteins that regulate transition through cell cycle phases are protooncogenes. Under unperturbed conditions, proto-oncogenes encode proteins required for cell growth, differentiation, and apoptosis. Certain alterations of proto-oncogenes result in their constituent expression; this is known as oncogene activation. Oncogene expression may confer growth advantage to that cell and increase its proliferation capacity, but it also leads to genome instability mediated by multifaceted interrelated mechanisms. Mechanisms that induce replication stress upon oncogene activation include the deregulation of CDK activity and overexpression of origin licensing factors, both of which alter origin firing. This also causes depletion of dNTPs and histones and generate regions of under-replicated DNA and ssDNA. Oncogene activation also alters transcriptional activity which increases the incidence of transcriptional-replication conflicts further expediting replication stress [38]. In this section, we have discussed some examples of oncogene mediated replication stress. Both Cyclins E and D complexes are required for progression through the G1 phase and the initiation of the S phase, respectively. Although Cyclin E-Cdk2 have multiple substrates, primarily the phosphorylation of Retinoblastoma (Rb) by Cyclin E and D releases it from the E2F transcription factor, the activity of which induces the expression of S-phase specific genes [40]. The overexpression of *Cyc E* in hyperplastic tissues causes perturbations of key steps during DNA replication. It increases firing from replication origins, DNA synthesis and reduces histone synthesis, and therefore depletes the overall pool of nucleotides and histone proteins, eventually leading to replicative stress and the deregulation of cell cycle progression [41]. The transcription of *Cyc E* during G1 is regulated by itself in an autoregulatory feedback mechanism. It is also integrated with the activity of another transcription factor Myc [42]. The overexpression of *c-Myc* has shown to increase the proportion of cells in S-phase. Increased levels of Myc protein therefore perturbs replication dynamics by increased asymmetric origin firing as well as uneven processivity on either side of the replication bubble, thus resulting in DNA damage due to fork stalling and collapse [43]. Asymmetric origin firing and replication forks are also induced by the activation and sustained stimulation of oncogenic RAS [44]. Oncogenic RAS can also directly interfere with ribonucleotide reductase (RRM2) which depletes the nucleotide pool and causes replication stress [45]. The RAS family of proto-oncoproteins comprises of small GTPases which function as signal transducers between cell surface receptors and intercellular proteins regulating cellular growth and many other physiological functions. Somatic mutations in RAS that induce their constitutive activation in turn promote cell proliferation, the suppression of apoptosis, and altered metabolism. Oncogenic RAS also alters cell cycle progression by deregulating ubiquitin mediated proteolysis which in turn affects the turnover rates of several proteins involved in cell proliferation and growth including Cyclin E, c-Myc, c-Jun, and Notch. Alterations in RAS expression are frequently observed in many cancers, mostly colorectal and lung adenocarcinomas [46,47,48]. The *CDC25* family of genes that encodes for the phosphatases is also critical for regulation of the cell cycle and checkpoint control. The overexpression of *CDC25* proteins causes aberrant replication initiation as they directly activate CDKs by dephosphorylation, resulting in premature cell cycle transitions. The sustained overexpression of *CDC25* results in the disruption of the checkpoints and chromosome aberrations [49]. A large number of oncogenes activated in a variety of cancers are associated with poor prognosis of the disease. Therefore, oncogene activation can be directly linked with replication stress-induced genome instability, which is central to genome alterations observed in cancer cells.

## 3. Genome Instability Due to Transcription and 3D Chromatin Structure

### 3.1. T–R Conflicts and Resolution of R Loops

For most genes, DNA replication and transcription are separated both spatially and temporally. On occasion, for genomic regions comprising of highly transcribed or very long genes, both transcription and replication can occur at the same time. This leads to an increased probability of collision between the transcription and replication machineries, also known as the T–R conflict. Apart from physical interaction of replisome and transcription machinery, the chromatin organization and the topology of actively transcribed regions can also interfere with DNA replication [50]. Two decades ago, it was demonstrated that higher transcription activity stimulates spontaneous mutations in yeast [51]. A similar approach in mammalian cells showed that transcriptionally active regions harbored a higher number of fragile sites and translocations compared to less active regions in the genome. It is possible that this is due to the torsional stress produced by super helical tension of negative supercoiling before and positive after the RNA polymerase complex. This type of torsional stress is relieved by Topoisomerases, and mutations or deficiency of these enzymes leads to trapped cleavage intermediates and persistent DNA damage that eventually converts to DSBs and short deletion [52]. T–R collisions result in unusual DNA structures and DNA lesions which are capable of eliciting ATM mediated DNA damage response [53]. 

Additionally, the regulation of gene expression also involves the wide-spread mechanism of RNA Polymerase II (RNAPII) pausing at proximal promoter regions prior to processive elongation. Recent studies reveal the enrichment of phosphorylated TRIM28 and γH2AX at the transcription start site of certain stimulus-inducible protein coding genes in humans. This indicates the involvement of induced DSBs, as well as ATM, DNA-PK, and Topoisomerase II (TopII) mediated DNA damage signaling in RNAPII pause release [54]. Therefore, transcription activity itself in addition to T–R conflicts can result in DNA damage and can be a potential source for genome instability. 

Multiple mechanisms are employed by the cell to resolve such unavoidable T–R conflicts and reduce their impact on genome integrity. *Escherichia coli* replisome has been demonstrated to either bypass, displace RNAP, or utilize the RNA as a primer for DNA synthesis during a co-directional T–R collision Co-directional T–R encounter is therefore considered as less damaging than head-on T–R collisions. [55]. It has been observed that in situations where replication and transcription cannot be separated spatially or temporally, cells generate a bias for co-directional progression of replication and transcription machineries. Recently, certain *cis* elements have been identified in mouse embryonic stem cells which possess enhancer-like properties and are involved in regulating the timing of replication initiation [56]. Consistent with this observation, replication origin firing near RNAPII-occupied transcription start sites of continuously transcribed genes has been demonstrated by Okazaki fragment sequencing. This analysis provides evidence for a bias towards the co-direction replication and transcription for certain genes [57]. Another mechanism has been observed in *Saccharamyces cerevisiae* and *Schizosaccharomyces Pombe,* where the 3′ termini of highly transcribed rDNA loci regions consisting of defined DNA sequences are tightly bound by certain non-nucleosomal proteins. These regions act as replication barriers and specifically prevent the progression of replication forks in the direction opposite to RNAP and thus generate a bias for the co-directional progression of replication and transcription [58]. 

Nevertheless, both head on and co-directional T–R conflict results in the formation of DNA–RNA hybrids or R-loops which pose a major obstacle to the replication forks [53]. R-loops also occur during cellular processes and perform important physiological functions. At the 3′UTR of genes, R-loops are involved in the termination of transcription and required for the release of mRNA from the DNA template. R-loops are present in CpG islands at the promoter regions genes where they prevent DNMT mediated gene silencing [59,60]. The R-loops once generated are very stable as RNA–DNA hybrids are thermodynamically more stable than duplex DNA [61,62]. If left unresolved or inefficiently processed, they pose significant impediments to both DNA replisome and transcription elongation further causing replication stress induced genotoxic stress

Both prokaryotes and eukaryotes employ multiple strategies for resolution of R-loops. These include enzymes from the THO/TREX complex, RNase H family, and helicase Sentaxin (SETX) [63,64,65]. Studies on *S. cerevisiae* showed that THO complex mutants exhibit increased formation of R-loops and replication perturbations. These mutant phenotypes were suppressed by the overexpression of RNase H. These studies suggest that the deficiency of THO complex activity in transcription is linked with the loss of genome stability [64]. Replication stress is also suppressed by SETX enzymes that are required for the dissolution of R-loops in both yeast and mammalian cells. The deficiency of RNAse H enzymes in mice leads to unresolved R-loops and the generation of ssDNA. This subsequently results in genome instability and embryonic lethality [66]. A member of the TREX complex known as DSS1 has been shown to physically interact with BRCA2. The interaction between DSS1 and BRCA2 is required for the resolution of R-loops as deficiency of either DSS1 or BRCA2 in cells caused the increased accumulation of R-loops. This study suggests a role of BRCA2 in R-loop processing which may be independent of its function in DNA repair and replication stress response [67]. A genome wide screen for factors that induced phosphorylation of H2AX, an established early mark of DNA damage identified proteins involved in RNA processing such as components of the spliceosome assembly and other splicing factors. Hence, the importance of RNA processing complexes in the maintenance of genome integrity cannot be undermined [68]. Oncogenic activation increases new origin firing. Replication forks emanating from these origins are more prone to T–R conflicts. Thus, unresolved R-loops serve as a major sources of genome instability [69]. 

### 3.2. Role of Transcriptional Activity in Replication Initiation 

The ORC proteins are capable of binding to the origin both in the euchromatin or the heterochromatin, however, the efficient loading of the MCM complex requires a more active or open state of the chromatin [70]. Acetylation Histone 4 and chromatin decondensation at the pre-RC have been shown to occur prior to the loading of the MCM 2-7 helicase complex. These processes involve the recruitment of histone acetyltransferase (HAT) enzymes such as HBO1 and several chromatin modifiers to the pre-RC. Chromatin remodelers such as Snf2 regulate origin licensing by regulating the binding of CDT1 and also promoting MCM2-7 association with DNA [71]. 

Recent studies have uncovered the role of histone methylases in replication progression and genome stability. Alteration of such epigenetic modifications are observed in response to hydroxyurea (HU) induced replication stress. These primarily include elevated levels of H3 lysine 4 trimethylation (H3K4Me3), which have also been associated with late replication origins that normally remain suppressed in the presence of replication stress. The histone methyl transferase MLL1 that induces H3K4 trimethylation, fluctuates during the cell cycle and its levels are stabilized by ATR in response to replication stress. The regulation of MLL1 levels is also involved in intra-S-phase checkpoint and constitutes a crucial component of the replication stress response. Cells with depleted MLL1 are not capable of preventing origin firing and suppressing DNA replication in response to replication stress [72]. Similarly, H3K79 methylation induced by DOT1L is involved in the activation of DNA damage checkpoint and the prevention of re-replication [73].

KMT2C and KMT2D-dependent H3K4 methylation is observed at replication forks in response to replication stress [74]. The cancer genome sequencing studies have identified these two histone methyltransferases to physically interact with the helicase RECQL5 [75]. These interactions are required for preventing T–R conflicts. Mutations in KMT2C and KMT2D have been found to be oncogenic which have the potential to induce DNA breaks and chromosomal rearrangements. These studies suggest that the regulation of chromatin state is linked with the location as well as the timing of origin firing and replication progression. These regulatory mechanisms involve a complex interplay between chromatin modeling and epigenetic modifications. Defects in these mechanisms are potential endogenous sources of genome instability.

### 3.3. Impact of 3D-Chromatin Organization on DNA Repair

The repair of different DNA lesions occurs in the context of a dynamic chromatin. The 3D organization of the chromatin is tightly regulated and restricted in the nuclear space. How chromatin organization impacts the process the DNA repair has been addressed by several studies [8]. Every chromosome occupies a specific location known as the chromosome territory (CTs) in the nucleus of the mammalian cell [76]. It has been shown that chromatin organization is non-random and regulated by gene density, such that the gene dense regions occupy the center of the nucleus while the gene sparse regions are located near the nuclear periphery [77,78]. Studies have shown that the activation of DNA damage response (DDR) leads to the relocation as well as the repositioning of CTs which may be dependent upon sensing and repair of the breaks. It has also been shown that chromatin domains from different CTs which comprise of multiple DSBs can relocate over long range distance of several microns into a cluster together into a repair center [78]. The relocation of the CT is reversible so they return to their previously occupied position upon the completion of the DSB repair [79].

The HR mediated DNA repair requires a homologous sequence which is used as the template for DNA synthesis at the break site. This process requires physical contact between the template and the DNA at the site of the break. During S/G2 phase such homologous sequence is present in the sister chromatid and therefore, available in the proximity of the DSB. On the contrary, in G1 phase when the sister chromatin is not present, DNA repair by HR may utilize a homologous sequence which from a different location in the genome, and this process can potentially result in structural rearrangements of the chromosome [78]. Therefore, CT relocation provides a means through which the cell coordinates global level chromatin remodeling with regulation of transcription. This is required for efficient DDR and is crucial for the maintenance of the genome.

## 4. Catastropic Consequences of Illegitimate Integration of Cell-Free Chromatin

Illicit recombination between repetitive sequences perturbs the chromatin organization and results in chromosomal rearrangements which are a hallmark of cancer. Heterochromatin formation at repetitive sequences and the silencing of transposable elements are evolutionary measures which suppress recombination events. 

The pathological effects of circulating cell-free chromatin (cfCh) in the blood on healthy cells of different tissues are not clearly understood. Mittra et al. have demonstrated that cfCh particles isolated from blood of cancer patients or healthy volunteers are capable of transfecting a variety of cells in culture and are readily detected in the nuclei of the host cells [80]. Once in the host cell nucleus, these cfCh particles associate with host cell genome and evoke DNA-damage-repair-response (DDR) within the host cell. Further, the integration of cfCh into the host cell genome resulted in DNA damage including DSBs and the activation of apoptotic pathways. This phenomenon was observed when cfCh were injected intravenously into Balb/C mice. A treatment involving circulating cfCh with DNase I and anti-histone antibody was sufficient to abrogate their activity and integration in the host cell genome [80]. The results from these studies suggest that circulating cfCh are source of endogenous DNA damage and have implication in aging and cancer. 

In the context of cancer, there is increasing evidence that suggests that cfCh are released from dying cancer or non-cancerous cells. These are then able to integrate into the genomes of the healthy cells in their vicinity and are followed by illegitimate recombination events which eventually cause significant chromosomal rearrangements in the host cells. During homeostasis, apoptotic cells are cleared by the process of phagocytosis [81]. The genomic integration of DNA released by phagocytosed apoptotic bodies with that of the host genome has been previously reported which can also induce such recombination events [82]. The study by Bergsmedh et al. used rat fibroblast cells which were first transfected with expression vectors expressing either Human Ras or c-myc and then co-cultivated with *p53^−^*/*p53^−^* mouse cells. The transfer of DNA from apoptotic rat fibroblasts to mouse cells was observed which was mediated by phagocytosis of apoptotic bodies by the mouse cells. Further, the genomic integration of rat DNA into the mouse genome and subsequent chromosomal rearrangements were also detected by FISH. The genomic integration of phagocytosed DNA resulted in DNA damage other phenotypes associated with oncogenic transformation in *p53^−^*/*p53^−^* mouse cells [82].

A recent study by Mittra et al. showed that when dead Jurkat cells, which were first labeled with fluorescently tagged nucleotide analogue Bromodeoxyuridine (BrdU) were incubated with cells of NIH3T3 cell line for 6 h, significant levels of fluorescent intensities from the incorporated BrdU molecules were observed in NIH3T3 cells. Furthermore, the uptake of cfCh by these actively growing cells was inhibited in presence of chromatin degrading compounds in the media of NIH3T3 cells suggesting that live cells in the vicinity of the apoptotic cells are can up take cell free chromatin released into their microenvironment. These studies were further extended to investigate if the uptake of cfCh by live cells could be observed in vivo. In these experiments, BrdU labelled apoptotic cells were injected intravenously into mice. The fluorescent label was detected not only in the nuclei of cells present in the vicinity of the injection site but also in distant organs including the brain, lungs, and liver. These cells also exhibited substantial DNA damage and inflammation [83]. The results of both in vivo and in vitro studies reveal that cfCh released from dying cells can be taken up by surrounding healthy cells, where the integration of cfCh with the genome of host cell genome causes DNA damage and the activation of DNA damage response.

In context of the chromatin, the illegitimate integration of any cfCh will impact the organization of chromatin fiber in the host cells and in turn affect DNA repair. This phenomenon is bound to increase genome instability in the host genome and shape the mutation landscape required for neoplasia formation.

## 5. Conclusions

A deeper understanding of endogenous mechanisms that cause the loss of genome integrity and mutational signatures in specific cancer cells is essential. These will aid in the identification of novel biomarkers, the evaluation of treatment response, and prognosis outcomes. How some forms of genome instability are well tolerated in normal cells but same are associated with malignancy and poor prognosis in cancer cells remains to be understood. The interplay of signaling pathways, chromatin structure, epigenetic regulation, and other cellular processes influence gene expression and DNA damage responses. Defects and inefficiency in one or more of these processes has the potential to promote the malignant transformation of cells. Cells from different developmental contexts respond differently to genotoxic stress. Some cells with exacerbated DNA damage initiate apoptotic pathways while others evolve alternate error prone repair mechanism and tolerate DNA damage with increased mutation load [84]. Mutational signatures documented in different cancer cell genomes provide insights into identifying key pathways that are compromised during the malignant transformation of cells. These have improved our understanding of genome changes acquired by cancer cells which enable them to tolerate the mutation burden, the patient’s immune response, and resist different therapies. In turn, these specific features of cancer cells can enable us to selectively target them. The identification of synthetic lethal interactions can help in the development of novel therapeutic interventions. A classic example of this include tumors with mutations in BRCA1/2 gene that are specifically sensitive to Poly ADP-Ribose Polymerase (PARP) inhibition [85]. Intriguingly, tumors deficient in other HR components but not BRCA genes also exhibit similar phenotypes and are sensitive to PARP inhibition. Other novel synthetic lethal interactions can also be used to target different types of cancers. The discovery of the phenomenon of illegitimate integration of cell free chromatin released from dying cell into healthy cells has important implications in genome instability and disease progression in the context of cancer. This process can bring about diverse genomic alterations which underlie somatic mosaicism, malignant transformation, and inflammation, all of which are crucial to cancer etiology. An improved understanding of these phenomena will provide us with invaluable insights regarding cancer and metastasis, which will immensely help with the development of newer and more effective treatment strategies.

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
