# Peer review of "Origin of Genome Instability and Determinants of Mutational Landscape in Cancer Cells"

_genes, 2020, doi:10.3390/genes11091101_

Round 1

Reviewer 1 Report

Manuscript ID: genes-917488

Title: Origin of genome instability and determinants of mutational landscape in cancer cells Authors: Sonam Mehrotra and Indraneel Mittra

Comments:

In this  manuscript, the authors  discuss recent insights about how some cellular processes, such as replication stress, oncogene activation, chromatin dynamics, 3D-chromatin organization and integration of illegitimate cfCh, contribute to the evolution of cancer cells.

The review is well written and provide a good overview of the current state of the research field.

However, in the last decade, numerous manuscripts have described new evidences regarding molecular mechanisms through which single- and double-strand breaks (some of which are also scheduled by the cells) can occur during transcription initiation and release of the paused and the elongation state. These DNA breaks, tightly associated with the transcription and not depending from the replication process, have also been proven linked to genome instability.

As the authors have only partial address this field mentioning to the role of the Topoisomerases (p6, l249-256), I strongly encourage them to improve this topic.

Minor point:

there are mistakes to be removed or amended:

  • P2, l78-86
  • P3, l138 “…..from by…” ???

Author Response

Response Reviewer 1 Comments:

Point 1: In this manuscript, the authors discuss recent insights about how some cellular processes, such as replication stress, oncogene activation, chromatin dynamics, 3D-chromatin organization and integration of illegitimate cfCh, contribute to the evolution of cancer cells.

The review is well written and provide a good overview of the current state of the research field.

Response 1: We thank the reviewer for this feedback. Based on the comments and suggestions, we have rewritten some of the sections to improve the manuscript.

Point 2: However, in the last decade, numerous manuscripts have described new evidences regarding molecular mechanisms through which single- and double-strand breaks (some of which are also scheduled by the cells) can occur during transcription initiation and release of the paused and the elongation state. These DNA breaks, tightly associated with the transcription and not depending from the replication process, have also been proven linked to genome instability.

As the authors have only partial address this field mentioning to the role of the Topoisomerases (p6, l249-256), I strongly encourage them to improve this topic.

Response 2: Considering the comments and suggestions by the reviewer, we have rewritten section 3.1 (lines 229 to 268). Here we have described the mechanisms that lead to unavoidable Transcription-Replication (T-R) conflicts. We have also referred to studies that have elucidated the role of DNA breaks and DNA damage response in RNA polymerase II pause release and elongation and provide evidence that transcription activity itself can be a potential source of genome instability.

This has been followed by a description of general strategies employed by cells to prevent or reduce the effects of T-R conflicts on genome integrity.  These include generation of a bias for co-directional T-R progression, prevention of head on T-R encounters and role of various protein complexes involved in resolution of R-loops.

 Point 3:

Minor point:

there are mistakes to be removed or amended:

  • P2, l78-86

Response 3: This paragraph has been removed.

  • P3, l138 “…..from by…” ???

Response 4: The sentence mentioned by the reviewer has been corrected.

Reviewer 2 Report

In this review, Mehrotra and Mittra discuss aspects and mechanisms of genome instability and consequences leading to cancer.

Although this is an interesting and timely review, the manuscript needs further improvement before acceptance. Aside of need of extensive language editing, there are a number of inaccurate statements (e.g. lines 94-95: replication stress is in fact a state and consequence of genome instability/mutations but not a “process”). Another example is the phrase: “While mutations in DNA repair genes are frequently associated with hereditary 34 cancers, the mutational signatures of sporadic cancers remain poorly defined.” (lines 34-35) is not correct. In fact, the high throughput sequencing revealed the mutational signatures of cancers. The authors need to indicate precisely their statements. This is also true for definitions, e.g. line 25: “Genome instability is a characteristic feature observed in most human cancers. It can be defined as elevated rate of mutations which over time leads to chromosomal instability and predisposes cells towards malignancy.” Basically, genome instability is the same as chromosomal instability, unless the authors wish to indicate some other ideas here.

The whole review seems to be a little bit unstructured. There are parts where there are connections missing to the general subject and/ or the sentences before. The reader is left with the question what this information has to with the paragraph above.

I suggest that the authors carefully go through the manuscript and state more precisely their arguments and better the structure the manuscript. For example, the introduction would benefit from a better organization.

Original structure:

Genome instability, Mutations that lead to genome instability, DSBs repair mechanisms, DNA damage response, DNA repair and mutations

Restructured:

Genome instability, Mutations that lead to genome instability (no need to make a difference between sporadic and hereditary cancer here), DNA damage response, DSB repair mechanisms

Comments to 2.3:

Why starting here right away with examples? Where is the general mechanism?

The authors jump from one example to the other without really finishing until the final consequences of the overexpression of the oncogene.

Comments to 3.1:

There is no explanation of the general mechanism of T-R conflict prevention

Author Response

Response to Reviewer 2 Comments:

In this review, Mehrotra and Mittra discuss aspects and mechanisms of genome instability and consequences leading to cancer.

Point 1:  Although this is an interesting and timely review, the manuscript needs further improvement before acceptance. Aside of need of extensive language editing, there are a number of inaccurate statements:

Response 1: Considering the reviewer’s comments, we have carefully revised and rewritten many parts of the manuscript to correct any inaccuracies.

 Point 2: e.g. lines 94-95: replication stress is in fact a state and consequence of genome instability/mutations but not a “process”.

Response 2: In agreement with the reviewer’s comment, and the statements in written in lines 94-95 (now 96-99 in the revised version) have been rephrased.

Point 3: Another example is the phrase: “While mutations in DNA repair genes are frequently associated with hereditary 34 cancers, the mutational signatures of sporadic cancers remain poorly defined.” (lines 34-35) is not correct. In fact, the high throughput sequencing revealed the mutational signatures of cancers.

Response 3:  We have revised lines 29-41 and included a brief summary of the recent high throughput sequencing study by the Pan-Cancer Analysis of Whole Genome (PNAWG) Consortium.

Point 4: The authors need to indicate precisely their statements. This is also true for definitions, e.g. line 25: “Genome instability is a characteristic feature observed in most human cancers. It can be defined as elevated rate of mutations which over time leads to chromosomal instability and predisposes cells towards malignancy.” Basically, genome instability is the same as chromosomal instability, unless the authors wish to indicate some other ideas here.

Response 4: To improve clarity, lines 25-27 have been rewritten. Here we want to suggest that accumulation of different genetic alterations ranging from single nucleotide change to chromosome rearrangements can predispose a cell towards malignancy.

Point 5: The whole review seems to be a little bit unstructured. There are parts where there are connections missing to the general subject and/ or the sentences before. The reader is left with the question what this information has to with the paragraph above.

I suggest that the authors carefully go through the manuscript and state more precisely their arguments and better the structure the manuscript. For example, the introduction would benefit from a better organization.

Original structure:

Genome instability, Mutations that lead to genome instability, DSBs repair mechanisms, DNA damage response, DNA repair and mutations

Restructured:

Genome instability, Mutations that lead to genome instability (no need to make a difference between sporadic and hereditary cancer here), DNA damage response, DSB repair mechanisms

Response 5: Based on reviewer’s feedback, we have carefully studied and revised the manuscript by rewriting several sections. These are indicated with track changes in the text. The Introduction has been restructured based on the suggestions provided by the reviewer.  All the statements indicated by the reviewer have been revised or rewritten.

Point 6: Comments to 2.3:

Why starting here right away with examples? Where is the general mechanism?

The authors jump from one example to the other without really finishing until the final consequences of the overexpression of the oncogene.

Response 6:  Considering reviewer’s feedback, a description of different mechanisms lead to replication stress upon oncogene activation has been included in the beginning of this section. This is then followed by citing examples of specific oncogenes and their role in deregulation of cell cycle resulting in replication stress.

Point 7: Comments to 3.1:

There is no explanation of the general mechanism of T-R conflict prevention

Response 7: Considering the comments and suggestions by the reviewer, we have rewritten section 3.1 (lines 229 to 268). In the revised version of this section, we have first described the mechanisms that lead to unavoidable Transcription-Replication (T-R) conflicts. We have also referred to studies that have elucidated the role of DNA breaks and DNA damage response in RNA polymerase II pause release and elongation which provide evidence for a link between transcription activity and genome instability. This is followed by a description of strategies employed by cells to prevent or reduce the effects of T-R conflicts on genome integrity.  These include generation of a bias for co-directional T-R progression, prevention of head on T-R encounters and role of various protein complexes involved in resolution of R-loops.

Reviewer 3 Report

In their review “Origin of genome instability and
determinants of mutational landscape in cancer cells”
Mehrotra and Mittra give an outline of the currently known
sources and causes for genome damage. Overall, they
describe how disturbances especially on the process of
replication can interfere with genome instability.
Additionally, they present the research of several of their own
studies wherein it has been shown that cell-free chromatin
can have deleterious effects when taken up by cells and
integrated into the genome.
In my opinion, the authors give a good overview across the
topic. However, I have a few questions and comments to
statements made throughout the review:
- lines 34-35 – “While mutations in DNA repair genes
are frequently associated with hereditary cancers, the
mutational signatures of sporadic cancers remain
poorly defined” It would be worth referencing some
of the work from Mike Stratton’s group (most
recently Alexandrov et al., Nature, 2020 578 94-101)
which actually does define many of the signatures.
- line 103 – Holliday junction-like (capitalized and
spelt differently, named after Robin Holliday)
- line 218 – “Active transcription occurs in the G1
phase” This is a surprising statement, and it is unclear
what “active” means. Different genes are transcribed
at various stages of the cell cycle (see for example, Liu
et al. PNAS 2017, 114 (13), 3473). Perhaps the authors
can cite a reference showing transcription is most
elevated in the G1 stage.
- line 227 – “As R-loop are more stable than duplex
DNA” Is this true? Can the authors provide a
reference?
- lines 253-254 – “super helical tension of negative
supercoiling before and after the RNA polymerase
3
complex” This is incorrect. Negative supercoiling
accumulates before the RNA Polymerase complex,
but positive supercoiling builds up after the RNA
Polymerase.
- lines 314 & 338 – For the sake of transparency, the
authors should be clear that these are their study.
Replacing “it has been demonstrated” with “we have
demonstrated” and replacing “a recent study…”
with “our recent study” would be sufficient.
Overall, I think the review gives a good outlook across
the different sources of replication stress and I can
recommend it, especially if the comments I made are
considered. Apart from that I have noticed some typos or
unclear use of words:
- At the end of the introduction, there still is the
explanation of how an introduction should be written
as according to the journal’s guidelines.
- In line 49 “…mechanisms which coordinate…”
(remove the s)
- In line 62-63 “DNA damage responses are…”
- In line 117 it should be “…accelerate…” (remove the
s)
- In line 133 “…can be re-initiated by bypassing…”
(remove “from”)
- In line 370 “…enable them…” (“them” instead of
“then”)
- In line 374 “…tumours deficient for other…” (remove
“in”)

Author Response

Response to Reviewer 3 Comments:

In their review “Origin of genome instability and determinants of mutational landscape in cancer cells”

Mehrotra and Mittra give an outline of the currently known sources and causes for genome damage. Overall, they describe how disturbances especially on the process of replication can interfere with genome instability.

Additionally, they present the research of several of their own studies wherein it has been shown that cell-free chromatin can have deleterious effects when taken up by cells and integrated into the genome.

In my opinion, the authors give a good overview across the topic. However, I have a few questions and comments to statements made throughout the review:

We thank the reviewer for the comments and hope that we have been able to improved the manuscript with the revisions.

Point 1:  lines 34-35 – “While mutations in DNA repair genes are frequently associated with hereditary cancers, the mutational signatures of sporadic cancers remain poorly defined” It would be worth referencing some of the work from Mike Stratton’s group (most recently Alexandrov et al., Nature, 2020 578 94-101) which actually does define many of the signatures.

Response 1:  Based on the reviewer's comments, we have substantially revised this section of the manuscript by modifying lines 29-41. We have included a brief summary of the recent high through put sequencing study by the Pan-Cancer Analysis of Whole Genome (PNAWG) Consortium  which  systematically characterized a repertoire of mutational signatures and their contribution to development if different human cancers and cited (Aleandrov et  al  Nature 2020).

Point 2: line 103 – Holliday junction-like (capitalized and spelt differently, named after Robin Holliday)

Response 2: We thank the reviewer for pointing out this error, which has been corrected. 

Point 3: line 218 – “Active transcription occurs in the G1 phase” This is a surprising statement, and it is unclear what “active” means. Different genes are transcribed at various stages of the cell cycle (see for example, Liu et al. PNAS 2017, 114 (13), 3473). Perhaps the authors can cite a reference showing transcription is most elevated in the G1 stage.

Response 3: Considering the comments and suggestions by the reviewer, we have rewritten section 3.1 (which are lines 229 to 268 in the revised version). The previous statement “active transcription occurs in the G1 phase has been removed”. Instead, in the revised version, we have described the mechanisms that lead to unavoidable Transcription-Replication (T-R) conflicts. We have also referred to studies that have elucidated the role of DNA breaks and DNA damage response in RNA polymerase II pause release and elongation. These provide evidence for a link between transcription activity and genome instability. This is followed by a description of the strategies employed by cells in order to prevent or reduce the effects of T-R conflicts on genome integrity.  These include generation of a bias for co-directional T-R progression, prevention of head on T-R encounters and the role of various protein complexes involved in resolution of R-loops.

Point 4:  line 227 – “As R-loop are more stable than duplex DNA” Is this true? Can the authors provide a reference?

Response 4: We have cited references for studies which provide evidence to suggest that R- loops or RNA/DNA hybrids are thermodynamically more stable that duplex DNA. (Roberts RW and Crothers 1992 Science; Skourti-Stathaki, K., N.J. Proudfoot Genes Dev 2020).

Point 5: lines 253-254 – “super helical tension of negative supercoiling before and after the RNA polymerase complex” This is incorrect. Negative supercoiling accumulates before the RNA Polymerase complex, but positive supercoiling builds up after the RNA Polymerase.

Response 5: Considering the comments from the reviewer, we have modified these statements (lines 237- 241) and appropriate references have been cited.

Point 6:  lines 314 & 338 – For the sake of transparency, the authors should be clear that these are their study. Replacing “it has been demonstrated” with “we have demonstrated” and replacing “a recent study…” with “our recent study” would be sufficient.

Response 6: As the studies mentioned here are by only one of the authors, therefore, we have replaced “it has been demonstrated” with “Mittra et al have demonstrated” and “a recent study…” with “recent study by Mittra et al” respectively.

Point 7: Overall, I think the review gives a good outlook across the different sources of replication stress and I can recommend it, especially if the comments I made are considered. Apart from that I have noticed some typos or unclear use of words:

- At the end of the introduction, there still is the explanation of how an introduction should be written as according to the journal’s guidelines.

Response 7: We thank the reviewer for pointing out these typos and other errors. We have carefully studied and manuscript and corrected these and other error in the manuscript.

 -In line 49 “…mechanisms which coordinate…” (remove the s)

Now in line 64

-In line 62-63 “DNA damage responses are…” 

Now in lines 51- 53. The statement has be rephrased.

-In line 117 it should be “…accelerate…” (remove the s)

Now in line 121

-In line 133 “…can be re-initiated by bypassing…” (remove “from”)

Now in line 136

-In line 370 “…enable them…” (“them” instead of “then”)

Now in line 410

- In line 374 “…tumours deficient for other…” (remove “in”)

 Now in line 414